# How Maternal Trauma Exposure Contributed to Children’s Depressive Symptoms following the Wenchuan Earthquake: A Multiple Mediation Model Study

**DOI:** 10.3390/ijerph192416881

**Published:** 2022-12-15

**Authors:** Yiming Liang, Yiming Zhao, Yueyue Zhou, Zhengkui Liu

**Affiliations:** 1Shanghai Key Laboratory of Mental Health and Psychological Crisis Intervention, Affiliated Mental Health Center (ECNU), School of Psychology and Cognitive Science, East China Normal University, Shanghai 200062, China; 2School of Psychology, Henan University, Kaifeng 475004, China; 3CAS Key Laboratory of Mental Health, Institute of Psychology, Chinese Academy of Sciences, Beijing 100101, China

**Keywords:** maternal depression, newborns, children, posttraumatic growth, Wenchuan earthquake

## Abstract

Although well-established literature has indicated the burden of mental health among victims after the Wenchuan earthquake, no research has focused on the mental health of mothers and their children who experienced the earthquake and were pregnant during or shortly after it. This study investigates the relationship between maternal trauma exposure (TE) and children’s depressive symptoms after the Wenchuan earthquake and explores the risk and protective factors underlying this relationship. A sample of 547 mother-child dyads, in which the mother experienced the Wenchuan earthquake, was used to assess maternal depressive symptoms, maternal TE, children’s depressive symptoms, children’s perceived impact of the earthquake and maternal posttraumatic growth (PTG). The results showed that maternal TE had two significant one-step indirect associations with children’s depressive symptoms (through children’s perceived impact of the earthquake and maternal PTG) and one two-step indirect association with children’s depressive symptoms (through maternal depressive symptoms via children’s perceived impact of the earthquake). The results indicated that maternal depressive symptoms, children’s perceived impact of the earthquake and maternal PTG mediated the association between maternal TE and children’s depressive symptoms. These findings highlight the importance of mothers in supporting the mental health of these children. Maternal depressive symptoms and PTG, two posttraumatic outcomes, played positive and negative roles in the intergenerational transmission of trauma. Thus, post-disaster interventions should reduce the maternal transmission of trauma-related information and improve maternal PTG to support children’s mental health.

## 1. Introduction

On 12 May 2008, the Wenchuan earthquake struck Sichuan Province in China, causing thousands of deaths and injuries and tremendous property loss. Earthquakes are one of the most destructive natural disasters, and survivors may suffer from a range of psychological problems. Prior studies on the Wenchuan earthquake have reported on the prevalence of posttraumatic stress disorder (PTSD), depression and anxiety among survivors [1,2,3,4,5]. These problems are common in pregnant women who have experienced an earthquake, and they have adverse effects on the birth outcomes of the women’s offspring [6]. In studies focusing on women who were pregnant during or shortly after the Wenchuan earthquake, the prevalence rates of PTSD and major depression were 12.2% and 40.8%, respectively, at one year after the earthquake [7], and the prevalence rate of anxiety was 52% at one month after the earthquake [7,8]. Previous research has found that prenatal exposure to an earthquake was correlated with lower birth weight, increased risk of preterm birth and lower gestational age [9,10,11], which may hinder both the emotional and cognitive development of the offspring [12,13,14]. Zacher et al. [15] found maternal exposure to Hurricane Katrina had a significant indirect effect on child mental health via maternal distress. Additionally, King et al. [16] found that maternal subjective distress following an ice storm had a significant effect on their children’s internalizing of problems (such as anxiety, depression and social withdrawal) and externalizing of problems (such as aggression and destructiveness). Similarly, Walder et al. [17] found that prenatal exposure to an ice storm predicted higher autism spectrum disorder symptoms in offspring. Until now, no research has paid attention to the mental health of newborns after the Wenchuan earthquake. The causes and consequences of their post-disaster mental health problems require additional consideration and systematic research.

An increasing amount of literature has focused on the intergenerational effects of trauma, especially when transferred from mothers to their children. The trauma exposure (TE) of mothers may result in psychopathology symptoms in their offspring [18]. As articulated in theories of intergenerational effects of trauma, trauma can be transmitted through both TE and mothers’ negative psychological responses [19]. Although traumatic events clearly have effects on the next generation, offspring maladaptation may not be the only consequence [20]. Robinson et al. [21] demonstrated maternal TE to be associated with the psychiatric symptoms of children, whereas other research has observed maternal TE to be positively associated with the well-being of offspring and negatively associated with maternal psychological maltreatment [22,23]. Therefore, the current study used mother-child paired data to explore the intergenerational effects of trauma and to identify potential risk and protective factors during this process. In the present study, mothers experienced the Wenchuan earthquake and were pregnant either during or shortly after the earthquake.

Depression is the most common mental health outcome of a natural disaster, and depressive symptoms may continue for many years after a disaster [24]. Guo, He, Qu, Wang and Liu [2] found that 8 years after the Wenchuan earthquake, 24.8% of adult survivors had probable depression. Moreover, compared to other post-disaster psychological consequences (e.g., PTSD), depression had a higher prevalence and was more chronic among traumatized populations after an earthquake [25]. Therefore, the survivors’ depressive symptoms for years after the disaster remain of significant concern.

One of the consistent predictors of post-disaster depression is TE [26]. Maternal TE from the Wenchuan earthquake is an important risk factor for the post-disaster depressive symptoms of mothers [27]. Children born after the earthquake did not witness the earthquake, but their perceived impact of this disaster may have similar psychological effects as the TE [28]. The most important impacts of the earthquake were the economic loss and the psychological symptoms among survivors [29]; therefore, we consider children’s perceived impact of the earthquake on their families, including economic and psychological outcomes. Children may learn from their mothers how their families were affected by the earthquake. Therefore, we hypothesized that maternal TE positively predicts maternal depressive symptoms and children’s perceived impact of the earthquake. However, we did not expect a direct association between maternal TE and children’s depressive symptoms because results regarding the relationship between maternal TE and child outcomes have been inconsistent [30,31,32]. As Lehrner and Yehuda [20] asserted, parental maladjustment following trauma can consistently predict offspring psychopathology, rather than parental TE.

Maternal depressive symptoms are one of the most important risk factors for early-onset depression in offspring [33]. The Integrative Model contains four mechanisms underlying the relationship between maternal depression and offspring psychopathology, including heritability, innate dysfunctional neuroregulatory mechanisms, exposure to the negative and/or maladaptive cognitions, behaviors, affects of mothers and exposure to stressful environments [34]. A 2011 meta-analysis revealed that children of depressed mothers were at higher risk of internalizing problems, including depression [35]. In addition, previous studies have shown that depression in children with depressed parents has an earlier onset, a more chronic course and an elevated rate of recurrence [36,37,38]. Compared with children of nondepressed mothers, children exposed to maternal depression have higher negative emotionality and lower self-concept, which may make them vulnerable to depression [39,40]. Thus, we focused on the effect of maternal depressive symptoms on children’s depressive symptoms and hypothesized that the former are positively associated with the latter.

Identifying potent risk factors for depression between mothers and children is critical for offering psychological intervention to families. Although the relationship between maternal depression and child depression has been well documented, the mechanism underlying this relationship in the context of trauma requires more exploration. Due to the negative bias, depressed mothers may recall more traumatic information and convey more traumatic memories to their children [21]. In other words, the offspring of depressed mothers may perceive more of their families’ traumatic exposure. Howard [28] found that children who learned about the trauma of family members exhibited higher rates of trauma symptomatology than children who did not experience such TE. Parents are the primary support source [41]. Children rely on their caregivers to keep them safe; thus, what the mother considers threatening and detrimental may have similar effects for her children [42]. Therefore, we hypothesized that children’s perceived impact of an earthquake on their families mediates the link between the depressive symptoms of mothers and children.

Despite the detrimental impact of TE on mental health, trauma survivors may also experience positive psychological changes, which is conceptualized as posttraumatic growth (PTG) [43]. As articulated in Tedeschi and Calhoun’s theory [44], PTG occurs when individuals’ beliefs and goals are reconstructed due to a traumatic event. The positive changes are thought to cover five domains, including new possibilities, relating to others, personal strength, appreciation of life and spiritual change. Current research indicates that PTG is common in people who have experienced traumatic events. For example, Linley and Joseph [45] found that after traumatic events, approximately 70% of individuals experienced positive psychological changes in at least one aspect. Jin et al. [46] found that 51.1% of the Wenchuan earthquake survivors reported experiencing different levels of PTG. Moreover, considerable research has shown that PTG is predicted by both objective TE (objective characteristics of the traumatic event) and subjective TE (immediate reactions, e.g., helplessness, a sense of loss of control and a feeling of life threats) [47]. Helgeson et al. [48] analyzed 87 studies and found that objective trauma severity was positively associated with growth and benefit finding. In addition, a prior study of the Wenchuan earthquake revealed that exposure severity was positively correlated with PTG for adolescent survivors [49]. The association between TE and PTG can be explained by the functional-descriptive model [50] and the conservation of resources (COR) theory [51]. According to Janoff-Bulman’s [52] functional-descriptive model, traumatic experience is a seismic challenge to the pre-trauma schema of individuals. This challenge shatters people’s beliefs about themselves, their social relationships and the world [50]. The breakdown of worldviews subsequently forces deliberate rumination, facilitates the reconfiguration of a more adaptive schema and creates meaning from the traumatic event [53]. A certain extent of TE is even necessary to engender PTG because transformation in worldviews requires a struggle with trauma [54]. Moreover, according to the COR theory, after a traumatic event, people experience the loss of resources and the risk of losing resources again [55]. They will seek new resources to compensate for this loss [51]. The new resources may include strengthened social relationships and social support, which is one of the aspects of PTG. Thus, we postulate that higher TE will predict higher PTG. However, the literature has largely regarded PTG as an outcome of other predictors. Little research has explored how the positive effects of PTG may be transmitted. Existing results on the relationship between parental PTG and child outcomes are complicated. In a sample of children who had siblings with a complex chronic health condition, Stephenson et al. [56] found that maternal PTG was correlated with fewer parent-reported child behavior problems. However, when child behavior problems were self-reported, only PTG in terms of relationships was correlated with fewer self-reported child behavior problems. Moreover, Schepers et al. [57] found that parental growth had an indirect effect on youth adaptation via attachment among childhood cancer survivors, whereas no direct effect was found. Thus, mothers with higher PTG may experience stronger relationships with their children and exhibit more positive parenting, such as providing emotional support and demonstrating warmth. These positive parenting behaviors have been proven to buffer against child psychopathic traits [58]. Therefore, we hypothesized that maternal PTG mediates the relationship between maternal TE and child depressive symptoms.

The current study explores the effects of maternal TE on children’s depressive symptoms and potential mechanisms among survivors of the Wenchuan earthquake. In particular, we hypothesized that (1) children’s perceived impact of the earthquake would mediate the relationship between maternal TE and children’s depressive symptoms; (2) maternal depressive symptoms would mediate the relationship between maternal TE and children’s perceived impact of the earthquake, such that greater maternal TE would predict more severe maternal depressive symptoms, which in turn would predict higher levels of children’s perceived impacts of the earthquake; and (3) maternal PTG would mediate the relationship between maternal TE and children’s depressive symptoms.

## 2. Methods

### 2.1. Procedures and Participants

The study was conducted in 2018 (10 years after the 2008 Wenchuan earthquake) in Beichuan County, which was most severely affected by the earthquake. This study focused on the mental health of mothers who were pregnant with their children during or shortly after the earthquake, as well as post-earthquake neonates. We recruited (1) mothers who had experienced the Wenchuan earthquake and (2) their children born between 12 May 2008, and 31 December 2009, who were attending Beichuan County’s largest primary school (the Yongchang primary school).

After explaining the study procedures to the mothers and children, we obtained informed consent from the mothers for them and their children to participate. Both the mothers and children were clearly informed that their participation was completely voluntary and that they were free to withdraw from the survey at any time, and all their information would remain confidential. The children were evaluated collectively in a group format during class by two postgraduates who were uniformly trained by using the same standardized instructions and tips. In addition, the children’s class teachers were also present. Before the children completed the questionnaire, they were again informed that their participation was completely voluntary and that they could withdraw from the survey at any time, and all their information would remain confidential. The mothers finished the questionnaires at home and returned them to our team the next day. All of the participants were told that psychologists in our team were available to provide psychological services if needed. The children received stationery as compensation for their participation after completing the questionnaire. The study design and procedures were approved by the ethics review committee of the Institute of Psychology, Chinese Academy of Sciences.

Seven hundred and seven pairs of mothers and children met our inclusion criteria in the Yongchang primary school. Among them, 558 pairs of mothers and children participated in this study (response rate: 78.93%). Because 11 children’s data were invalid (too many missing values or careless answers), we generated 547 pairs of matching data (from both the children and mothers), which were included in the final data analysis. The ultimate sample consisted of 547 children (253 boys and 294 girls; mean age = 9.05 years, *SD* = 0.64) and their mothers (mean age = 37.09 years, *SD* = 6.35). A total of 269 children (49.18%) belonged to the Han ethnicity group, 255 children (46.62%) belonged to the Chiang ethnicity group, 14 children (2.56%) belonged to other ethnicity groups, and 9 children (1.64%) did not respond regarding their ethnicity. The demographic information of the mothers and categories of TEs are presented in Table 1.

### 2.2. Measures

#### 2.2.1. Maternal Depressive Symptoms

The Chinese version [59] of the Beck Depression Inventory (BDI) [60] was used to measure maternal depressive symptoms. Participants responded using a four-point Likert scale ranging from 0 to 3 for all 21 items, with higher scores indicating higher levels of depressive symptoms. Previous literature suggests that total scores ≥14 indicate moderate-to-severe depressive symptoms [61]. In the present study, the scale exhibited good internal consistency (Cronbach’s α of 0.89).

#### 2.2.2. Maternal Posttraumatic Growth

Maternal PTG was measured by the 21-item self-report Posttraumatic Growth Inventory [43]. This scale consisted of five parts: relating to others (a sample item was “I learned a great deal about how wonderful people are”), new possibilities (a sample item was “I established a new path for my life”), personal strength (a sample item was “I discovered that I’m stronger than I thought I was”), spiritual change (a sample item was “A better understanding of spiritual matters”) and appreciation of life (a sample item was “An appreciation for the value of my own life”). For each item, participants responded on a 6-point scale ranging from 0 to 5 (0 = not at all, 5 = very much). Total scores ≥ 63 were considered to indicate moderate PTG [62]. In the present study, the Cronbach’s α for the scale was 0.94.

#### 2.2.3. Maternal Trauma Exposure

Seven trauma exposure questions were created to assess maternal TE to the Wenchuan earthquake, such as whether the respondents had “been trapped”, “been injured”, “lost a relative” or “witnessed buildings falling” during it. Each item used true/false (1/0 scoring) response options. A general score for trauma exposure was computed by summing these item scores.

#### 2.2.4. Children’s Depressive Symptoms

The Chinese version [63] of the Children’s Depression Inventory (CDI) [64] was employed to assess depressive symptoms among children. The scale consisted of 27 items and included five subscales: anhedonia (including items on sleep disturbance, fatigue, loneliness, etc.), negative mood (including items on sadness, pessimistic worrying, self-blame, etc.), negative self-esteem (including items on pessimism, self-hate, suicidal ideation, etc.), ineffectiveness (including items on self-deprecation, school work difficulty, school performance decrement and low self-esteem) and interpersonal problems (including items on misbehavior, social withdrawal, disobedience and fighting). All the items were scored on a three-point scale ranging from 0 to 2, with higher scores indicating more severe depressive symptoms. A cutoff for the total score equal to 19 or higher is adequate for the general screening of depression in children [65]. The Cronbach’s α of the scale was 0.85.

#### 2.2.5. Children’s Perceived Impact of the Earthquake on the Family

Two items were created to assess children’s perceived impact of the Wenchuan earthquake on their families, which covered psychological and economic effects. The two items were “According to your feelings, to what extent did the Wenchuan earthquake affect your family’s economic situation” and “According to your feelings, to what extent did the Wenchuan earthquake affect your family members’ psychological state”. The children responded to each item using a five-point Likert scale ranging from 1 to 5 (1 = mild, 5 = severe). A general score for the child’s perceived impact of the earthquake on their family was computed by summing the two item scores.

### 2.3. Data Analysis

An analysis of missing data was first conducted. The item-level data were missing for 3.75% across all the variables. To assess the pattern of missing data, Little’s missing data completely at random test was conducted. The results revealed that data were missing in a random way [χ2 (59) = 65.258, *p* = 0.268]. Missing data were handled using the maximum likelihood procedure.

Descriptive analyses and Pearson correlations were conducted in IBM SPSS (Version 22.0 for Windows). Descriptive analyses were calculated for all the measures. As some variables did not satisfy normality, we performed logarithmic transformation. Then, Pearson’s correlations were conducted to explore the associations among the variables that were of the most concern.

We then constructed a path analysis using Mplus version 8.3 [66] to examine the following models. First, a direct model was assessed to explore the impact of maternal TE on children’s depressive symptoms. Second, children’s perceived impact of the earthquake on their family, maternal depressive symptoms and maternal PTG were inserted into the direct model as mediators. We then added predictive paths from maternal depressive symptoms to children’s perceived impact of the earthquake on their family and maternal PTG to children’s perceived impact of the earthquake on their family. Moreover, because depressive symptoms and PTG are two related posttraumatic outcomes [67], a correlation between maternal PTG and maternal depressive symptoms was also established in the model. Mother’s age and educational level were included as control variables in the model. Finally, we added a multiple indirect effects model (see Figure 1). In addition, we conducted bias-corrected bootstrap tests (*n* = 1000 bootstrap samples) with a 95% confidence interval (CI) to test the significance of the indirect effects. Bootstrapping is useful for testing the significance of indirect effects and can provide accurate estimates of CIs, which are based on an examination of the empirical distribution of the indirect effect. Confidence limits are computed for the sampling distribution [68,69]. We evaluated the model fit using the following goodness-of-model-fit indexes: comparative fit index (CFI), Tucker–Lewis index (TLI), standardized root mean square residual (SRMR), root mean square error of approximation (RMSEA) and χ^2^/df. Cutoffs with CFI > 0.90, TLI > 0.90, SRMR < 0.08, RMSEA < 0.06 and χ^2^/df < 5 were applied for model evaluation [70].

## 3. Results

### 3.1. Descriptive Statistics and Correlations among the Main Measures

The average BDI score (maternal depressive symptoms) was 8.96 (*SD* = 8.57), and 24.76% of the mothers met the cutoff for BDI. The average CDI score (children’s depressive symptoms) was 12.51 (*SD* = 7.61), and 21.48% of the children met the cutoff for CDI. The average PTG score was 65.33 (*SD* = 19.88), and 56.31% met the cutoff for PTG. The means, standard deviations and correlations for the main measures are shown in Table 2. The correlation between maternal TE and children’s perceived impact of the earthquake on their family was positively significant (*r* = 0.22, *p* < 0.01). Maternal depressive symptoms were positively associated with maternal TE (*r* = 0.26, *p* < 0.01) and children’s perceived impact of the earthquake on their family (*r* = 0.17, *p* < 0.01). Children’s depressive symptoms were positively associated with children’s perceived impact of the earthquake on their family (*r* = 0.16, *p* < 0.01) and maternal depressive symptoms (*r* = 0.11, *p* < 0.05), but were not significantly associated with maternal TE (*r* = −0.01, *p* = 0.83). Maternal PTG was significantly associated with maternal TE (*r* = 0.19, *p* < 0.01) and children’s depressive symptoms (*r* = −0.09, *p* < 0.05), but not significantly associated with children’s perceived impact of the earthquake on their family (*r* = 0.05, *p* = 0.25) or with maternal depressive symptoms (*r* = −0.03, *p* = 0.50).

### 3.2. Examination of Multiple Mediating Effects

First, a direct effects model was constructed and indicated a direct association between maternal TE and children’s depressive symptoms. Path analyses revealed that maternal TE did not have a significant direct relation with children’s depressive symptoms (β = −0.04, *p* = 0.401).

Second, a final multiple indirect effects model was established according to the procedures described in the data analysis (see Figure 1). This model showed a good fit: CFI = 0.995, TLI = 0.978, SRMR = 0.017, RMSEA = 0.017, and χ^2^/df = 1.147. The results showed that maternal TE had two significant one-step indirect associations with children’s depressive symptoms via children’s perceived impact of the earthquake on their family and maternal PTG. Maternal TE also had a significant two-step indirect association with children’s depressive symptoms via maternal depressive symptoms and through children’s perceived impact of the earthquake on their family. However, the one-step indirect association from maternal TE to children’s depressive symptoms, through maternal depressive symptoms, and the two-step indirect association from maternal TE to children’s depressive symptoms, through maternal PTG via children’s perceived impact of the earthquake on their family, were not significant. In addition, maternal PTG was significantly negatively correlated with maternal depressive symptoms (*r* = − 0.09, *p* = 0.040).

Finally, we conducted bias-corrected bootstrap tests with a 95% CI to test the significance levels of indirect effects in the multiple indirect effects model. The results, presented in Table 3, indicated that three indirect paths from maternal TE to children’s depressive symptoms in the final model were significant. Two significant one-step indirect associations were maternal TE → children’s perceived impact → children’s depressive symptoms (standardized β = 0.031, *SE* = 0.011, *p* = 0.005, 95% CI = [0.047, 0.211]) and maternal TE → maternal PTG → children’s depressive symptoms (standardized β = −0.021, *SE* = 0.012, *p* = 0.047, 95% CI = [−0.198, −0.007]). One significant two-step indirect association was maternal TE → maternal depressive symptoms → children’s perceived impact → children’s depressive symptoms (standardized β = 0.005, *SE* = 0.003, *p* = 0.049, 95% CI = [0.002, 0.012]). In addition, a nonsignificant one-step indirect association was maternal TE → maternal depressive symptoms → children’s depressive symptoms (standardized β = 0.014, *SE* = 0.011, *p* = 0.198, 95% CI = [−0.009, 0.035]), and a nonsignificant two-step indirect association was maternal TE → maternal PTG → children’s perceived impact → children’s depressive symptoms (standardized β = −0.001, *SE* = 0.002, *p* = 0.801, 95% CI = [−0.003, 0.004]).

In summary, the multiple mediation model showed that (1) maternal TE did not have a direct effect on children’s depressive symptoms; (2) two paths in which maternal TE had a positive association with children’s depressive symptoms were: directly via children’s perceived impact of the earthquake and via maternal depressive symptoms and then through children’s perceived impact of the earthquake; and (3) maternal TE had a negative association with children’s depressive symptoms via maternal PTG. Moreover, in the multiple mediation model, maternal depressive symptoms were not significantly associated with children’s depressive symptoms, as this relationship was fully mediated by children’s perceived impact of the earthquake. Maternal PTG was not significantly associated with children’s perceived impact of the earthquake.

## 4. Discussion

The present study is the first to examine the relationship between maternal TE and newborns’ mental health after the Wenchuan earthquake. We found that maternal TE had two significant positive indirect associations with children’s depressive symptoms, including a one-step indirect association (through children’s perceived impact of the earthquake) and a two-step indirect association (through maternal depressive symptoms via children’s perceived impact of the earthquake). We also found that maternal TE had a significant negative indirect association through maternal PTG with children’s depressive symptoms. The findings show that two posttraumatic outcomes (maternal depressive symptoms and PTG) play important roles in the intergenerational transmission of trauma. Thus, mental health care for newborns is needed after the Wenchuan earthquake, especially for mothers who have a high level of depressive symptoms.

Maternal TE had two significant positive indirect associations with children’s depressive symptoms, which were mediated by maternal depressive symptoms and children’s perceived impact of the earthquake. Consistent with our hypothesis, we found that maternal depressive symptoms were positively associated with children’s depressive symptoms after the Wenchuan earthquake. This result is also consistent with the results of previous studies, which have indicated the intergenerational transmission of depression [71,72]. We found that following the Wenchuan earthquake, the children of mothers with more severe depressive symptoms were at higher risks of psychopathology. Furthermore, these findings suggested that mothers’ psychological responses played an important role in the post-disaster adjustments of children.

We also found that the relationship between maternal and child depressive symptoms was mediated by the children’s perceived impact of the earthquake, which supported our hypothesis. As articulated in Goodman and Gotlib’s model [34], family environmental factors may contribute to the intergenerational transmission of depression. The present study proved that this model can be applied to the context of natural disasters. Due to cognitive biases, depressed people favor negative information and recall more negative memories [73,74]. Depressed mothers may more frequently refer to the negative aspects of the Wenchuan earthquake and its detrimental impact on the entire family than nondepressed mothers. As a result, children may learn from their mothers to be fearful and worry about potential subsequent earthquakes, and this transmitted trauma can also cause psychological distress [75]. Although these children were not directly exposed to the earthquake, they may develop depressive symptoms similar to those of their mothers as a consequence of the intergenerational transmission of trauma. Because exposure to trauma-related information is a pathway for the transmission of depressive symptoms, post-disaster intervention can instruct mothers to restrict their negative references to the earthquake and to offer emotional support when children learn of the details of the earthquake.

Notably, the results also showed that the relationship between maternal TE and children’s depressive symptoms was mediated by a positive posttraumatic outcome (maternal PTG). A certain extent of TE may be essential to trigger cognitive reappraisal of the traumatic event, which engenders PTG [76]. Previous studies have found that a severe trauma experience and high PTG can coexist in the same individual [77,78]. However, there was no direct correlation between maternal TE and children’s depressive symptoms. This result is consistent with the results of previous studies that find no significant direct effect of parental TE on offspring psychopathology [79,80]. Compared with exposure to trauma, parental coping and adaptation following trauma may play a more important role in predicting child outcomes [20]. In the present study, the adverse impact of maternal TE might have been diminished due to mothers’ successful adjustment, which was indicated by high maternal PTG. While suffering from distress caused by the earthquake, mothers may also experience positive changes in their perceptions of life and relationships. Parental past trauma is bound to affect the offspring, but whether children develop vulnerability or resilience largely depends on the responses of their parents [20]. If a mother fails to recover from trauma and develops depressive symptoms, her children are also more likely to experience depression due to the intergenerational transmission of trauma, as previously noted. However, when the mother succeeds in finding new meaning in the adverse experience, learning to appreciate life and experiencing stronger connections to family members, the children’s appraisal and understanding of the earthquake can be improved, which buffers against the detrimental impact of exposure to trauma-related information. Thus, maternal exposure to the Wenchuan earthquake may not necessarily cause psychopathology in children. This result highlights a direction for future clinical practice. Postdisaster intervention can pay more attention to improving the mothers’ understanding of the disaster and instructing them to find positive aspects from the adverse trauma experience. We also found that maternal PTG was not a predictor of children’s perceived impact of the earthquake. This result is not surprising because the earthquake’s perceived economic influence is objective trauma, which may be independent of the psychological states of mothers. Moreover, we measured the children’s perceived psychological impacts of the earthquake on the entire family. Mothers’ positive changes may not have a significant psychological effect on the entire family.

Despite its novel contributions, the present study has limitations. First, we could not draw causal conclusions due to the cross-sectional design. Second, self-report questionnaires were used, which might cause reporting bias. There is a need for replication of our findings in clinically diagnosed samples via structured interviews. Moreover, we examined several important variables related to children’s depressive symptoms, but there are likely other potential mechanisms underlying the transmission of these symptoms. For example, negative parenting behaviors can also mediate maternal and offspring depression [81]. Hughes et al. [82] found that maternal depressive symptoms were correlated with children’s depressive symptoms via insecure attachment. We suggest that future efforts explore the effects of other potential factors after natural disasters. Finally, the prevalence of PTSD after a disaster is often low due to the disappearance of trauma cues [83], and this study was conducted 10 years after the Wenchuan earthquake. Therefore, in the current study, we chose depression as an indicator of mothers’ post-disaster psychological outcomes instead of PTSD. However, the effect of PTSD on the intergenerational transmission of trauma needs to be further investigated in future studies.

## 5. Conclusions

The current study contributes to the field by being the first to delineate the pathways between maternal TE and newborns’ mental health after the Wenchuan earthquake. Our findings suggest several theoretical implications. First, this study reveals the potential mechanisms underlying the intergenerational transmission of trauma among earthquake survivors. While maternal post-disaster TE can be transmitted to children via maternal depressive symptoms and children’s perceived impact of the earthquake, maternal TE may not necessarily cause adverse child outcomes. Maternal PTG can serve as a protective factor against the transmission of trauma.

The current study also has significant clinical implications. The mental health of newborns after a major disaster requires more attention from clinicians. Given that mothers play an important role in the palliation of children’s depressive symptoms, interventions aimed at reducing maternal transmission of trauma-related information and enhancing maternal PTG may be effective in supporting children’s mental health. Specifically, mothers can be instructed to view the traumatic event in a different way, learn to appreciate the meaning of life and seek social connections with others, all of which may help facilitate resilience in both mothers and children who are prone to post-disaster maladjustment.

## Figures and Tables

**Figure 1 ijerph-19-16881-f001:**
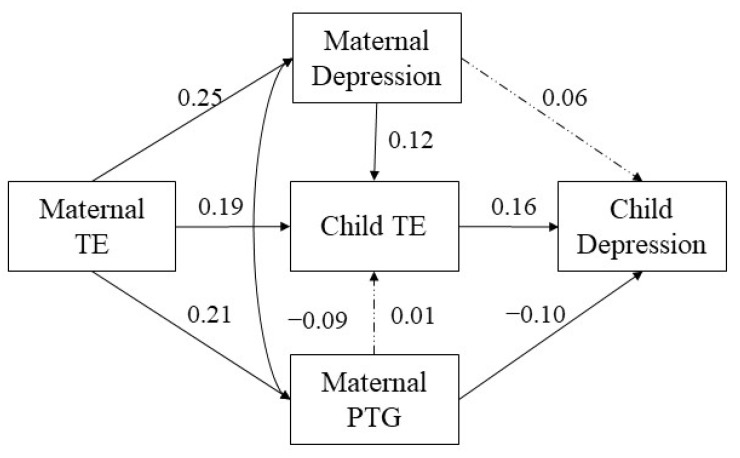
Multiple indirect effect model. Note. Standardized coefficients are reported. Solid lines represent significant paths, and the dashed line represents an insignificant path. TE: trauma exposure; children’s perceived impact: children’s perceived impact of the earthquake on their family; PTG: posttraumatic growth.

**Table 1 ijerph-19-16881-t001:** Mothers’ demographic and trauma exposure information (*n* = 547).

	Category	*n* (Percentage)
Education level	Never went to school	6 (1.10%)
	Primary school	98 (17.92%)
	Junior high school	245 (44.79%)
	High school	122 (22.30%)
	College and above	66 (12.07%)
Ethnicity	Han	336 (61.43%)
	Chiang	195 (35.65%)
	Other	14 (2.56%)
Marital status	First marriage	425 (77.70%)
	Divorced	22 (4.02%)
	Remarried	34 (6.22%)
	Widowed	4 (0.73%)
	Other	51 (9.32%)
Perceived household economic level	Low	33 (6.03%)
	Relatively low	69 (12.61%)
	Average	357 (65.27)
	Relatively high	58 (10.60%)
	High	3 (0.55%)
Employment status	Full-time	277 (50.64%)
	Part-time	64 (11.70%)
	Unemployed	159 (29.07%)
Trauma exposure	Was trapped	138 (25.23%)
	Was injured	108 (19.74%)
	Witnessed a building collapse	411 (75.14%)
	Saw a dead body	161 (29.43%)
	Lost one (or more) relative(s)	240 (43.88%)
	Was terrified or helpless	467 (85.37%)
	Witnessed people dying	192 (35.10%)

Note. The total number for some variables did not reach 547 because of missing data.

**Table 2 ijerph-19-16881-t002:** Means, standard deviations, and correlations among the study variables (*n* = 547).

	1	2	3	4	Mean (*SD*)
1. Maternal TE	–				2.99 (1.99)
2. Children’s perceived earthquake impact	0.22 **	–			5.31 (2.56)
3. Maternal depressive symptoms	0.26 **	0.17 **	–		8.96 (8.57)
4. Children’s depressive symptoms	−0.01	0.16 **	0.11 *	–	12.51 (7.61)
5. Maternal PTG	0.19 **	0.05	−0.03	−0.09 *	65.33 (19.88)

Note. * *p* < 0.05, ** *p* < 0.01. Children’s perceived earthquake impact: children’s perceived impact of the earthquake on their family; PTG: posttraumatic growth.

**Table 3 ijerph-19-16881-t003:** Standardized regression coefficients and 95% confidence intervals (CIs) for the indirect effects.

Paths	Standardized β	95% CI
One-step mediation		
maternal TE → maternal depressive symptoms → children’s depressive symptoms	0.014	[−0.009, 0.035]
maternal TE → children’s perceived earthquake impact → children’s depressive symptoms	0.031	[0.047, 0.211]
maternal TE → maternal PTG → children’s depressive symptoms	−0.021	[−0.198, −0.007]
Two-step mediation		
maternal TE → maternal depressive symptoms → children’s perceived earthquake impact → children’s depressive symptoms	0.005	[0.002, 0.012]
maternal TE → maternal PTG → children’s perceived earthquake impact → children’s depressive symptoms	−0.001	[−0.003, 0.004]

Note. TE: trauma exposure; children’s perceived earthquake impact: children’s perceived impact of the earthquake on their family; PTG: posttraumatic growth.

## Data Availability

The data presented in this study are available on request from the corresponding author.

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
