# Peer review of "How Maternal Trauma Exposure Contributed to Children’s Depressive Symptoms following the Wenchuan Earthquake: A Multiple Mediation Model Study"

_ijerph, 2022, doi:10.3390/ijerph192416881_

Round 1

Reviewer 1 Report

I would like to thank you for the opportunity to review the article: “How maternal trauma exposure contributed to children’s depressive symptoms following the Wenchuan earthquake: A multiple mediation model study”. 

This is a very interesting research, dealing with a relevant topic and offering clinical hints for post-traumatic interventions with pregnant women exposed to natural disasters. The results of the multiple mediation model - highlighting the role played by maternal depressive symptoms, children’s perceived impact of the earthquake, and maternal post-traumatic growth in the association between maternal trauma and children’s depressive symptoms - significantly contributed to the understanding of the intergenerational transmission of trauma, in terms of potential risk and protective factors

The introduction is based on a rigorous and critical analysis of the most relevant literature in the field, with research gaps clearly identified. Each research question has been addressed with a proper methodology (in terms of assessment instruments, study design, and statistical analyses). With regard to the last point, a minor suggestion would be to include Standard Error (SE) and p-value indexes in the results section (Lines 349-360, p.9).

Moreover, results are clearly presented, and the discussion is well supported. 

Overall, the article meets the criteria for acceptance without any further revision’s request.

Author Response

Dear Reviewer #1,

Thank you very much for giving us the opportunity to revise our manuscript entitled “How maternal trauma exposure contributed to children’s depressive symptoms following the Wenchuan earthquake: A multiple mediation model study” (Manuscript Number: ijerph-2078501). We greatly appreciate your comments and suggestions. Your expertise was very helpful for us in improving the quality of the paper. We have revised the manuscript and would like to resubmit it for publication in the International Journal of Environmental Research and Public Health.

We hope that the revised manuscript addresses most of the concerns that you raised and incorporates most of your suggestions. As a result of your valuable suggestions, the quality of the manuscript has been significantly improved. We hope to receive a favorable response from you. Please find our point-by-point responses attached.

Responses to Reviewer #1

This is a very interesting research, dealing with a relevant topic and offering clinical hints for post-traumatic interventions with pregnant women exposed to natural disasters.  The results of the multiple mediation model - highlighting the role played by maternal depressive symptoms, children’s perceived impact of the earthquake, and maternal post-traumatic growth in the association between maternal trauma and children’s depressive symptoms - significantly contributed to the understanding of the intergenerational transmission of trauma, in terms of potential risk and protective factors.

The introduction is based on a rigorous and critical analysis of the most relevant literature in the field, with research gaps clearly identified.  Each research question has been addressed with a proper methodology (in terms of assessment instruments, study design, and statistical analyses).  With regard to the last point, a minor suggestion would be to include Standard Error (SE) and p-value indexes in the results section (Lines 349-360, p.9).

Response: Thank you for your careful reading and encouragement. We added SE and p value indexes in the paragraph.

(p. 9) Two significant one-step indirect associations were maternal TE→ children’s perceived impact→ children’s depressive symptoms (standardized β = 0.031, SE = 0.011, p = 0.005, 95% CI = [0.047, 0.211]) and maternal TE→ maternal PTG→ children’s depressive symptoms (standardized β = −0.021, SE = 0.012, p = 0.047, 95% CI = [−0.198, −0.007]). One significant two-step indirect association was maternal TE→ maternal depressive symptoms→ children’s perceived impact→ children’s depressive symptoms (standardized β = 0.005, SE = 0.003, p = 0.049, 95% CI = [0.002, 0.012]). In addition, a nonsignificant one-step indirect association was maternal TE→ maternal depressive symptoms→ children’s depressive symptoms (standardized β = 0.014, SE = 0.011, p = 0.198, 95% CI = [−0.009, 0.035]), and a nonsignificant two-step indirect association was maternal TE→ maternal PTG→ children’s perceived impact→ children’s depressive symptoms (standardized β = −0.001, SE = 0.002, p = 0.801, 95% CI = [−0.003, 0.004]).

Reviewer 2 Report

This is an observational study to investigate the relationship between maternal trauma exposure and children’s depressive symptoms after the Wenchuan earthquake and explores the potential mechanisms underlying this relationship. The results showed that maternal depressive symptoms, children’s perceived impact of the earthquake and maternal posttraumatic growth mediated the association between maternal trauma exposure and children’s depressive symptoms. The findings are new and important in this research area, although there are several issues to be addressed.

1.                   The first issue is that the representativeness of the target population of this study is unclear. The subjects of this study were indicated (1) mothers who had experienced the Wenchuan earthquake and (2) their children who were born between May 12, 2008, and December 31, 2009, who were attending Beichuan County’s largest primary school. However, there is no information on how many students attended the school, how many were surveyed, and what the response rate was. The authors should add the information in more detail.

2.                   The authors confirm the association between variables by Pearson's correlation coefficient, but have they confirmed the normality of the variables?

3.                   Maternal educational history, marital status, economic status, employment status, etc. seem to be related to child depression, have these variables been incorporated into the analysis?

4.                   The association between each variable in the path analysis does not appear to be very strong, but to what extent does it contribute to the child's depression?

Author Response

Dear Reviewer#2,

Thank you very much for giving us the opportunity to revise our manuscript entitled “How maternal trauma exposure contributed to children’s depressive symptoms following the Wenchuan earthquake: A multiple mediation model study” (Manuscript Number: ijerph-2078501). We greatly appreciate your comments and suggestions. Your expertise was very helpful for us in improving the quality of the paper. We have revised the manuscript and would like to resubmit it for publication in the International Journal of Environmental Research and Public Health.

We hope that the revised manuscript addresses most of the concerns that you raised and incorporates most of your suggestions. As a result of your valuable suggestions, the quality of the manuscript has been significantly improved. We hope to receive a favorable response from you. Please find our point-by-point responses attached.

Responses to Reviewer #2

This is an observational study to investigate the relationship between maternal trauma exposure and children’s depressive symptoms after the Wenchuan earthquake and explores the potential mechanisms underlying this relationship. The results showed that maternal depressive symptoms, children’s perceived impact of the earthquake and maternal posttraumatic growth mediated the association between maternal trauma exposure and children’s depressive symptoms. The findings are new and important in this research area, although there are several issues to be addressed.

Response: Thank you for your careful reading and encouragement. We appreciate your comments and have revised the manuscript accordingly. Please see the following point-by-point responses and the revised manuscript.

  1. The first issue is that the representativeness of the target population of this study is unclear. The subjects of this study were indicated (1) mothers who had experienced the Wenchuan earthquake and (2) their children who were born between May 12, 2008, and December 31, 2009, who were attending Beichuan County’s largest primary school. However, there is no information on how many students attended the school, how many were surveyed, and what the response rate was. The authors should add the information in more detail.

Response: Thank you for pointing out this issue. Seven hundred and seven pairs of mothers and children met our inclusion criteria in this school. Among them, 558 pairs of mothers and children participated in this study (response rate: 78.93%). Because 11 children’s data were invalid (too many missing values or careless answers), we generated 547 pairs of matching data (from both the children and mothers), which were included in the final data analysis. We described the information in more detail in the revised version.

(p. 5) Seven hundred and seven pairs of mothers and children met our inclusion criteria in the Yongchang primary school. Among them, 558 pairs of mothers and children participated in this study (response rate: 78.93%). Because 11 children’s data were invalid (too many missing values or careless answers), we generated 547 pairs of matching data (from both the children and mothers), which were included in the final data analysis.

  1. The authors confirm the association between variables by Pearson's correlation coefficient, but have they confirmed the normality of the variables?

Response: Thank you for pointing out this issue. A normality test was performed on the variables used to calculate the correlation analysis. Since some variables did not satisfy normality, we performed logarithmic transformation. Then, Pearson’s correlations were conducted. We updated the results in Table 2.

(p. 7) Since some variables did not satisfy normality, we performed logarithmic transformation. Then, Pearson’s correlations were conducted to explore the associations between the variables that were of the most concern.

  1. Maternal educational history, marital status, economic status, employment status, etc. seem to be related to child depression, have these variables been incorporated into the analysis?

Response: Thank you for pointing out this issue. We tried to add marital status, economic status, and employment status as control variables to the model, and the results showed that these variables were not correlated with the variables we focused on. Only the mother’s educational level was related to child depression. Thus, we added the mother’s educational level as a control variable to the model and updated the results of the new model.

(p. 7) Mother’s age and educational level were included as control variables in the model.

  1. The association between each variable in the path analysis does not appear to be very strong, but to what extent does it contribute to the child's depression?

Response: Thank you for pointing out this issue. We reported the standardized β of the significant paths in the results. In fact, the effects of these pathways on child depression were not strong. Therefore, we pointed out that future studies need to explore other potentially important factors in the relationship between maternal and offspring depression in the limitations section.

(p. 9) Two significant one-step indirect associations were maternal TE→ children’s perceived impact→ children’s depressive symptoms (standardized β = 0.031, SE = 0.011, p = 0.005, 95% CI = [0.047, 0.211]) and maternal TE→ maternal PTG→ children’s depressive symptoms (standardized β = −0.021, SE = 0.012, p = 0.047, 95% CI = [−0.198, −0.007]). One significant two-step indirect association was maternal TE→ maternal depressive symptoms→ children’s perceived impact→ children’s depressive symptoms (standardized β = 0.005, SE = 0.003, p = 0.049, 95% CI = [0.002, 0.012]). In addition, a nonsignificant one-step indirect association was maternal TE→ maternal depressive symptoms→ children’s depressive symptoms (standardized β = 0.014, SE = 0.011, p = 0.198, 95% CI = [−0.009, 0.035]), and a nonsignificant two-step indirect association was maternal TE→ maternal PTG→ children’s perceived impact→ children’s depressive symptoms (standardized β = −0.001, SE = 0.002, p = 0.801, 95% CI = [−0.003, 0.004]).

(p. 11) we examined several important variables related to children’s depressive symptoms, but there are likely other potential mechanisms underlying the transmission of these symptoms. For example, negative parenting behaviors can also mediate maternal and offspring depression [81]. Hughes et al. [82] found that maternal depressive symptoms were correlated with children’s depressive symptoms via insecure attachment. We suggest that future efforts explore the effects of other potential factors after natural disasters.

Reviewer 3 Report

The work presented is very interesting and innovative. It was a great pleasure to be part of your review.

I leave some suggestions in order to improve the work presented:

Abstract: Lines 18-19 – indicate what potential mechanisms are they studying. Keywords: insert one related to children’ variables.

Introduction: Overall, the main ideas are expressed. As a reader, I feel some ideas maybe repeated and some paragraphs are too big.

An example: line 74 “One important trigger of depression is trauma.” But the authors already begin to approach trauma. So, the idea seemed repeated.

Lines 53-54 “Additionally, King et al. (2012) found that maternal subjective distress following an ice storm had a significant effect on the internalizing and externalizing problems of children.” – an example could be interesting.

Although the authors spoke about Mother TE and its’ effects on children in the context of an earthquake. Are there other studies, regarding other type of punctual disaster, that focused on its effect on children?

Line 138: punctuation is missing.

Along the text, the authors identify the hypotheses. Since they describe them in the end of the introduction, I believe they could just highlight the idea in the text and leave the hypotheses formulation for the problematic.

Method: an item example by subdimension could be inserted.

Results: in general, the data is very well explored. Just two notations: why the children gender was not considered? Line 320: the sentence is missing a purpose.

Discussion: The study limitations and future research could be more explored. Line 387: I believe the word “first” is not needed.

Author Response

Dear Reviewer#3,

Thank you very much for giving us the opportunity to revise our manuscript entitled “How maternal trauma exposure contributed to children’s depressive symptoms following the Wenchuan earthquake: A multiple mediation model study” (Manuscript Number: ijerph-2078501). We greatly appreciate your comments and suggestions. Your expertise was very helpful for us in improving the quality of the paper. We have revised the manuscript and would like to resubmit it for publication in the International Journal of Environmental Research and Public Health.

We hope that the revised manuscript addresses most of the concerns that you raised and incorporates most of your suggestions. As a result of your valuable suggestions, the quality of the manuscript has been significantly improved. We hope to receive a favorable response from you. Please find our point-by-point responses attached.

Responses to Reviewer #3

The work presented is very interesting and innovative. It was a great pleasure to be part of your review. I leave some suggestions in order to improve the work presented:

Response: Thank you for your careful reading and encouragement. We appreciate your comments and have revised the manuscript accordingly. Please see the following point-by-point responses and the revised manuscript.

Abstract: Lines 18-19 – indicate what potential mechanisms are they studying. Keywords: insert one related to children’ variables.

Response: Thank you for pointing out this issue. We clarified this sentence. We inserted “children” in the keywords.

(p. 1) This study investigates the relationship between maternal trauma exposure (TE) and children’s depressive symptoms after the Wenchuan earthquake and explores the risk and protective factors underlying this relationship.

Introduction: Overall, the main ideas are expressed. As a reader, I feel some ideas maybe repeated and some paragraphs are too big. An example: line 74 “One important trigger of depression is trauma.” But the authors already begin to approach trauma. So, the idea seemed repeated.

Response: Thank you for your valuable advice. We deleted some repetitive sentences in the introduction. For example, “One important trigger of depression is trauma. According to the fifth edition of the Diagnostic and Statistical Manual of Mental Disorders (DSM-5), trauma is defined as exposure to actual or threatened death, serious injury or sexual violence. One of the most common types of traumatic events is natural disasters, including earthquakes, wildfires, floods and hurricanes (Liu et al., 2017).”

Lines 53-54 “Additionally, King et al. (2012) found that maternal subjective distress following an ice storm had a significant effect on the internalizing and externalizing problems of children.” – an example could be interesting.

Response: Thank you for your valuable advice. We added some examples.

(p. 2) King et al. [16] found that maternal subjective distress following an ice storm had a significant effect on the internalizing problems (such as anxiety, depression and social withdrawal) and externalizing problems (such as aggression and destructiveness) of children.

Although the authors spoke about Mother TE and its’ effects on children in the context of an earthquake. Are there other studies, regarding other type of punctual disaster, that focused on its effect on children?

Response: Thank you for pointing out this issue. We cited a study following Hurricane Katrina, which also found that maternal exposure to Hurricane Katrina had a significant indirect effect on child mental health via maternal distress.

(p. 2) Zacher et al. [15] found that maternal exposure to Hurricane Katrina had a significant indirect effect on child mental health via maternal distress.

Line 138: punctuation is missing.

Response: Thank you for careful reading. We added a punctuation.

Along the text, the authors identify the hypotheses. Since they describe them in the end of the introduction, I believe they could just highlight the idea in the text and leave the hypotheses formulation for the problematic.

Response: Thank you for your valuable advice. We deleted conclusions already drawn in the previous literature and only hypothesized what we most concerned in this study.

(p. 4) (1) children’s perceived impact of the earthquake would mediate the relationship between maternal TE and children’s depressive symptoms; 2) maternal depressive symptoms would mediate the relationship between maternal TE and children’s perceived impact of the earthquake, such that greater maternal TE would predict more severe maternal depressive symptoms, which in turn would predict higher levels of children’s perceived impacts of the earthquake; and 3) maternal PTG would mediate the relationship between maternal TE and children’s depressive symptoms.

Method: an item example by subdimension could be inserted.

Response: Thank you for your valuable advice. We added item examples by subdimension.

(p. 6) This scale consisted of five parts: relating to others (a sample item was “I learned a great deal about how wonderful people are”), new possibilities (a sample item was “I established a new path for my life”), personal strength (a sample item was “I discovered that I’m stronger than I thought I was”), spiritual change (a sample item was “A better understanding of spiritual matters”) and appreciation of life (a sample item was “An appreciation for the value of my own life”).

The scale consisted of 27 items and included five subscales: anhedonia (including items on sleep disturbance, fatigue, loneliness, etc. ), negative mood (including items on sadness, pessimistic worrying, self-blame, etc. ), negative self-esteem (including items on pessimism, self-hate, suicidal ideation, etc. ), ineffectiveness (including items on self-deprecation, school work difficulty, school performance decrement, and low self-esteem) and interpersonal problems (including items on misbehavior, social withdrawal, disobedience, and fighting).

Results: in general, the data is very well explored. Just two notations: why the children gender was not considered?

Response: Thank you for pointing out this issue. First, based on theory, child depression does not show gender differences until early adolescence. The children in this study were approximately 10 years old. Second, based on the data, children’s depression was not significantly different by gender (p =.114).

Line 320: the sentence is missing a purpose.

Response: Thank you for careful reading. The sentence in line 320 is a note of Table 2. We reformatted it.

Discussion: The study limitations and future research could be more explored.

Response: Thank you for pointing out this issue. For example, we pointed out that future studies need to explore other potentially important factors in the relationship between maternal and offspring depression in the limitations section.

(p. 11) we examined several important variables related to children’s depressive symptoms, but there are likely other potential mechanisms underlying the transmission of these symptoms. For example, negative parenting behaviors can also mediate maternal and offspring depression [81]. Hughes et al. [82] found that maternal depressive symptoms were correlated with children’s depressive symptoms via insecure attachment. We suggest that future efforts explore the effects of other potential factors after natural disasters.

Line 387: I believe the word “first” is not needed.

Response: Thank you for careful reading. We deleted it.
